# Encoding of social exploration by neural ensembles in the insular cortex

Isamu Miura[1,2☯], Masaaki Sato[1,3☯¤], Eric T. N. Overton[1], Nobuo Kunori[1,4], Junichi Nakai[3,5], Takakazu Kawamata[2], Nobuhiro Nakai[1,6], Toru Takumi[1,3,6]*

**1** RIKEN Brain Science Institute, Wako, Saitama, Japan, **2** Department of Neurosurgery, Tokyo Women's Medical University, Shinjuku, Tokyo, Japan, **3** Graduate School of Science and Engineering, Saitama University, Sakura, Saitama, Japan, **4** The National Institute of Advanced Industrial Science and Technology, Tsukuba, Ibaraki, Japan, **5** Division of Oral Physiology, Department of Oral Function and Morphology, Tohoku University Graduate School of Dentistry, Aoba, Sendai, Japan, **6** Department of Physiology and Cell Biology, Kobe University School of Medicine, Chuo, Kobe, Japan

☯ These authors contributed equally to this work.
¤ Current address: Department of Neuropharmacology, Hokkaido University Graduate School of Medicine, Kita, Sapporo, Japan
* takumit@med.kobe-u.ac.jp

**Data Availability Statement:** All relevant data are within the paper and its Supporting Information files.

**Funding:** This work was supported in part by KAKENHI (16H06316, 16H06463) from Japan

## Abstract

The insular cortex (IC) participates in diverse complex brain functions, including social function, yet their cellular bases remain to be fully understood. Using microendoscopic calcium imaging of the agranular insular cortex (AI) in mice interacting with freely moving and restrained social targets, we identified 2 subsets of AI neurons—a larger fraction of "Social-ON" cells and a smaller fraction of "Social-OFF" cells—that change their activity in opposite directions during social exploration. Social-ON cells included those that represented social investigation independent of location and consisted of multiple subsets, each of which was preferentially active during exploration under a particular behavioral state or with a particular target of physical contact. These results uncover a previously unknown function of AI neurons that may act to monitor the ongoing status of social exploration while an animal interacts with unfamiliar conspecifics.

## Introduction

Social animals communicate and interact with conspecifics both cooperatively and competitively for reproduction and survival. Social interaction between 2 individuals typically begins with an appetitive phase that reflects interest in peers, such as approaching and investigation by touching and sniffing, followed by a consummatory phase that expresses a repertoire of goal-directed social behaviors, such as aggression, mating, or parenting, according to age, gender, and familiarity of the conspecific [1,2]. Recent advances in social neuroscience in rodents have revealed the role of critical centers for social behavioral control such as the medial preoptic area of the hypothalamus [3, 4], the posterodorsal and posteroventral medial amygdala [5, 6], the ventrolateral subdivision of the ventromedial hypothalamus [7, 8], and the medial prefrontal cortex (mPFC) [9–13]. However, functional characterization of additional network

Society for the Promotion of Science (JSPS) (https://www.jsps.go.jp/english/e-grants/index. html), Intramural Research Grant for Neurological and Psychiatric Disorders of NCNP (https://www. ncnp.go.jp/en/), the Takeda Science Foundation (https://www.takeda-sci.or.jp/index.html) to TT, KAKENHI (17H05985, 19H04942) to MS, and KAKENHI (15H05723) to JN. The funders had no role in study design, data collection and analysis, decision to publish, or preparation of the manuscript.

**Competing interests:** The authors have declared that no competing interests exist.

**Abbreviations:** AAV, adeno-associated virus; AI, agranular insular cortex; ASD, autism spectrum disorder; CNMF, constrained non-negative matrix factorization; FOV, field of view; GCaMP6f, GFP-based Calcium Calmodulin Probe 6f; GRIN, gradient refractive index; HC, home cage; IC, insular cortex; IDPS, Inscopix Data Processing Software; LC, linear chamber; mPFC, medial prefrontal cortex; NND, nearest neighbor distance; ns, not significant; PBS, phosphate-buffered saline; PFA, paraformaldehyde.

nodes is essential for the full understanding of neural circuit mechanisms underlying complex social behavior.

The insular cortex (IC), which lies deep within the lateral sulcus in humans and on the lateral aspect of the neocortex in rodents, is involved in a wide variety of functions, including multisensory integration, interoception, outcome prediction, decision-making, salience and valence coding, self-awareness, and complex social and emotional processes, including empathy [14–24]. It forms an anatomical hub with reciprocal connections to sensory, emotional, motivational, and cognitive systems—including the sensory and frontal cortices, amygdala, thalamus, and nucleus accumbens—as well as with neuromodulatory inputs [25–28]. Functional magnetic resonance imaging studies in humans suggest that the IC serves as the core of a "salience network" that acts to detect novel and behaviorally relevant stimuli [16, 29]. Furthermore, hypoactivity and dysfunctional connectivity of the IC are hallmarks of individuals with autism spectrum disorder (ASD) [30, 31]. These findings imply that the IC constitutes a key node for the social brain network and plays a pivotal role in social cognition and behavior [18–20, 32]. In this study, we therefore sought to examine how neurons in the IC encode information regarding social behavior by visualizing neuronal dynamics using microendoscopic calcium imaging in freely moving, socially interacting mice. Our study particularly focused on the agranular insular cortex (AI), the most anterior part of the IC that lacks the granular layer 4 and has denser limbic connections than the rest of the IC [25, 28].

## Results

### Multiple subsets of insular cells encode social exploration behavior

To elucidate how AI neurons encode direct social interaction with unfamiliar conspecifics, we first conducted a home-cage (HC) experiment [33], in which a male mouse was allowed to explore a novel object placed in its HC for the first 4 min, followed by another 4 min of exploration of a freely moving male stranger mouse that was introduced to replace the object (Fig 1A). Subject mice displayed only occasional direct nasal contacts with the novel object during control sessions, although they often looked and sniffed remotely toward it (13.1 ± 6.5 times; total duration, 58.7 ± 22.5 s, mean ± SD, $n$ = 9 mice). In contrast, they showed highly frequent nasal contacts with the stranger mice during interaction sessions (Fig 1B and 1C). The subject mice spent 30.5% of the total time on social interaction, defined by their nasal contact to the stranger mice (Fig 1D). Furthermore, 48.6%, 45.1%, and 6.3% of the social interaction period was spent in contact with the nose, body, and anogenital area of the stranger mice (Fig 1C–1E). Besides contact initiated by the subject mice, the stranger mice also contacted the subject mice for 6.2% of the total time. The subject mice remained stationary for 70.0% of the total time, and the fraction of stationary periods increased to 82.3% when only the social interaction periods were considered.

We then sought AI neurons that exhibited social interaction-associated activity using microendoscopic calcium imaging in subject mice (Fig 2A–2C and S1 Movie). We calculated similarities between the vectors representing their binarized neuronal activity and those representing when the subject mouse interacted with the stranger mouse. In the analysis, we selected cells whose activity was significantly correlated or anticorrelated with the social interaction period but not with generic behavioral states such as either moving or stationary periods. Out of 737 cells from 9 mice, we identified 22.8% ("Social-ON cells," 168 cells) and 1.4% ("Social-OFF cell," 10 cells) of total cells whose activity was correlated and anticorrelated with social interaction period, respectively (Fig 2D and 2E). The average event rate of Social-ON cells during periods of no social interaction ("nonsocial period") was significantly lower than cells that were neither Social-ON nor Social-OFF cells ("nonsocial cells") and substantially

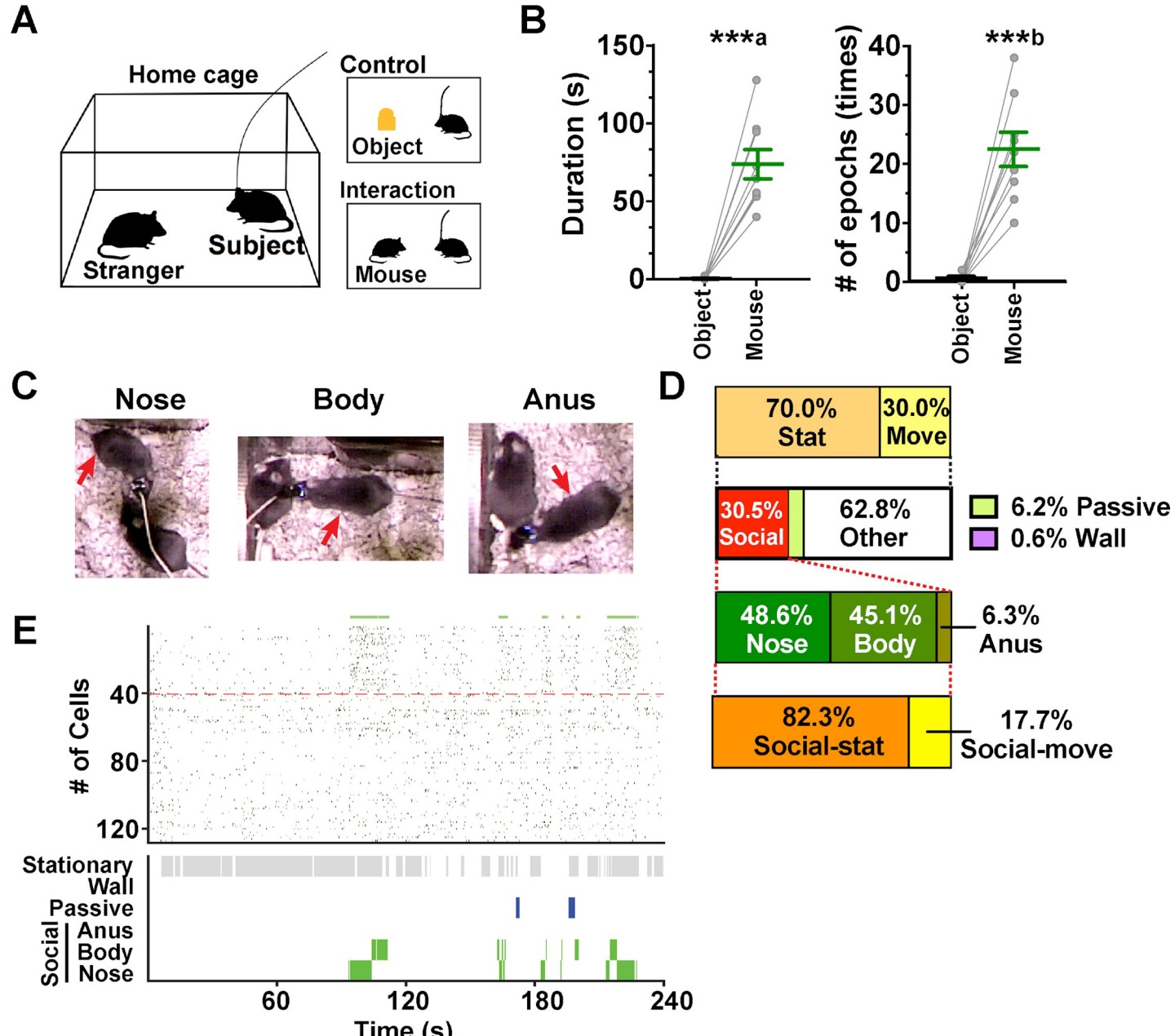

**Fig 1. Behavior during HC experiments.** (A) Experimental design of HC experiment. A subject mouse was allowed to explore a novel object in its HC for 4 min in a control session (top right), followed by an additional 4 min of an interaction session with a stranger mouse (bottom right). (B) Total duration (left) and the number of nasal contact epochs (right) with a novel object during control sessions and with a stranger mouse during interaction sessions. \*\*\*a, $P < 0.0001$ versus object, $t_{(8)} = 7.90$, paired $t$ test. \*\*\*b, $P < 0.0001$ versus object, $t_{(8)} = 7.35$, paired $t$ test, $n = 9$ mice (S1 Data, sheet Fig 1B). (C) Sample video frames of contact with the nose, body, and anus of the stranger. The subject mouse (red arrow) has a miniaturized microscope attached to its head. (D) Characterization of the behavior of subject mice in HC experiments. The fractions of time spent moving ("Move") and not moving ("Stat") and those for social interaction ("Social"), passive touch by the stranger ("Passive"), contact with the wall ("Wall"), and otherwise ("Other") in entire sessions are shown in the top and upper middle bars, respectively. The bars presented in the lower middle and bottom indicate the fractions of time spent touching the nose, body, and anogenital areas ("Anus") and those spent moving ("Social-move") and not moving ("Social-stat") during the periods of social interaction, respectively. (E) Behavior of a subject mouse during a single HC experiment (bottom). Nonmoving periods ("Stationary"), contact with the wall ("Wall"), passive touch by the stranger mouse ("Passive"), and contact with the anogenital area ("Anus"), body, and nose of the stranger are presented from top to bottom within the panel. The top panel presents a raster plot showing $Ca^{2+}$ events of a population of AI neurons imaged during the same experiment ($n = 128$ neurons; see Fig 2A–2C for technical details). The epochs of social interaction are indicated by green horizontal bars shown on the top. Social-ON cells are sorted above the red dashed line. AI, agranular insular cortex; HC, home cage.

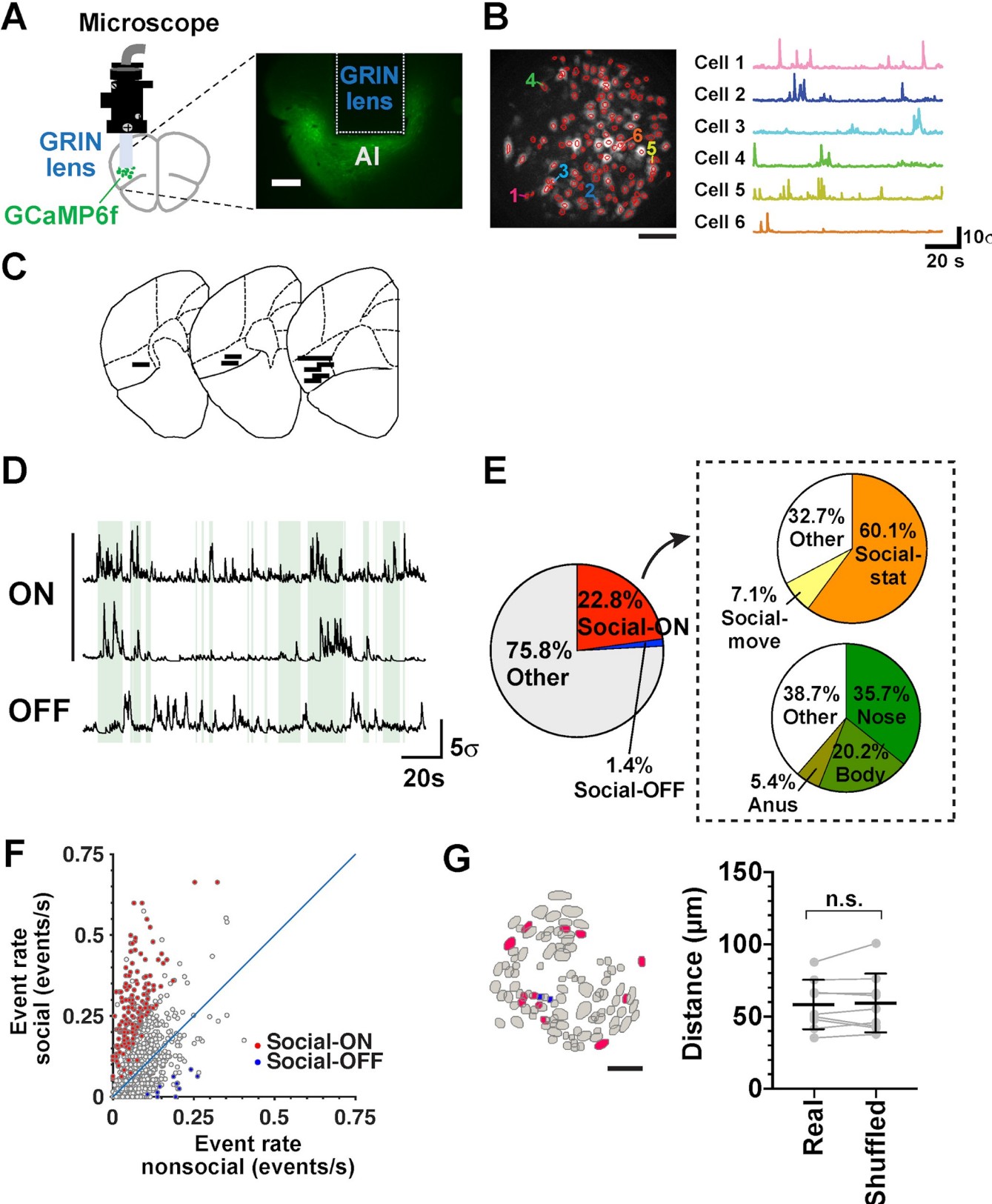

**Fig 2. Social-ON cells and Social-OFF cells in the AI.** (A) A schematic of microendoscopic imaging (left). AI neurons virally transfected with GCaMP6f were imaged using a miniaturized head-mounted microscope through a chronically implanted GRIN lens. A fluorescence image of a coronal section shows the trace

of a GRIN lens implanted close to GCaMP6f-expressing AI neurons (right). Scale bar = 200 μm. (B) A FOV showing contours of a population of imaged AI neurons (left, scale bar = 100 μm). Six example neurons are numbered, and their time-varying GCaMP6f fluorescence changes are shown on the right side. (C) Reconstructed positions of the implanted GRIN lenses in the AI ($n$ = 9 mice). Approximate centers of the lenses along the anteroposterior axis are shown by black horizontal lines in drawings of representative coronal sections modified from the Allen Mouse Brain Atlas. (D) GCaMP ΔF/F traces of 2 Social-ON cells and a Social-OFF cell. The epochs of social interaction are indicated in green. (E) A pie chart showing the fractions of each category of cells in the total cell population (left, $n$ = 737 cells from 9 mice). "Other" in the left chart includes cells whose activity was either correlated with passive touch by the stranger (0.9% of total cells), overall moving periods (2.3% of total cells), or overall stationary periods (2.4% of total cells). The pie charts shown within the dashed rectangle demonstrate the fractions of each subcategory of cells in the Social-ON cell population ($n$ = 168 cells). (F) A scatter plot of $Ca^{2+}$ event rates of individual neurons during social interaction and nonsocial periods ($n$ = 737 cells from 9 mice). The blue line indicates equivalence. Social-ON cells ($n$ = 168 cells) and Social-OFF cells ($n$ = 10 cells) are shown in red and blue, respectively (S1 Data, sheet Fig 2F). (G) Anatomical distribution of social cells in a FOV (left, $n$ = 106 cells). Social-ON cells ($n$ = 12 cells) and Social-OFF cells ($n$ = 2 cells) are shown in red and blue, respectively. Scale bar = 100 μm. Average NNDs between Social-ON cells calculated using real ("Real") and shuffled data ("Shuffled") are shown in the right panel ($P$ = 0.59, real versus shuffled, $t_{(8)}$ = 0.55, paired $t$ test, $n$ = 9 mice; S1 Data, sheet Fig 2G). AI, agranular insular cortex; FOV, field of view; GCaMP6f, GFP-based Calcium Calmodulin Probe 6f; GRIN, gradient refractive index; NND, nearest neighbor distance.

increased during the social interaction period compared to the nonsocial period (S1A Fig and Fig 2F). In contrast, the average event rate of Social-OFF cells during the nonsocial period was significantly higher than that of the nonsocial cells and markedly decreased during social interaction compared to the nonsocial period (S1A Fig and Fig 2F). As a result, activity of Social-ON cells and Social-OFF cells exhibited strong biases toward the social interaction period and the nonsocial period (Social-ON cells, 0.625 ± 0.167, $n$ = 168 cells; Social-OFF cells, −0.735 ± 0.188, $n$ = 10 cells; mean ± SD), respectively (S1B Fig). Within the microendoscopic field of view (FOV), Social-ON cells and Social-OFF cells coexisted in an intermingled manner (Fig 2G; see also S6A Fig). The nearest neighbor distances (NNDs) between Social-ON cells calculated using real and shuffled data did not differ significantly, indicating that Social-ON cells are distributed randomly within the AI (Fig 2G).

Interestingly, 60.1% (101 cells) and 7.1% (12 cells) of Social-ON cells also exhibited activity significantly correlated with the parts of social interaction periods during which the subject mouse was stationary and moving, respectively (termed "social-stationary cells" and "social-movement cells"; Fig 2E and S2A Fig). These subcategories of cells demonstrated higher event rates during the relevant behavioral state only when the subject mouse was socially interacting (S2B Fig), indicating that subsets of insular Social-ON cells are preferentially activated while a subject mouse interacts with a stranger under a particular behavioral state. Finally, we examined whether specific social contact by the subject mice also influenced the activity of Social-ON cells. We found that 35.7% (60 cells), 20.2% (34 cells), and 5.4% (9 cells) of Social-ON cells showed activity that was significantly correlated with the period during which they had contact with the nose, body, and anogenital area of the stranger mice (Fig 2E and S2C and S2D Fig), although the amount of time spent in contact with the anus was low ($n$ = 4 out of 9 mice; range 0.7%–7.3% of total time; Fig 1D). In summary, these findings suggest that insular Social-ON cells do not merely respond to social interaction in general but also represent information regarding behavioral state and the target of contact during social interaction.

## Location-independent coding of social investigation by AI neurons

We next investigated activity of AI neurons in a linear-chamber (LC) apparatus, a modified version of the three-chamber test that can control spatial factors and physical contact (Fig 3A) [34]. In these experiments, male subjects' preference for social stimuli was assessed by comparing the amount of time spent investigating 2 separate small chambers that contained either a male stranger mouse or a novel inanimate object. The investigation behavior was defined as poking of the subject mouse's nose into holes of the acrylic wall of the small chambers. The transparency of the wall and its holes allowed transmission of olfactory, visual, auditory, and

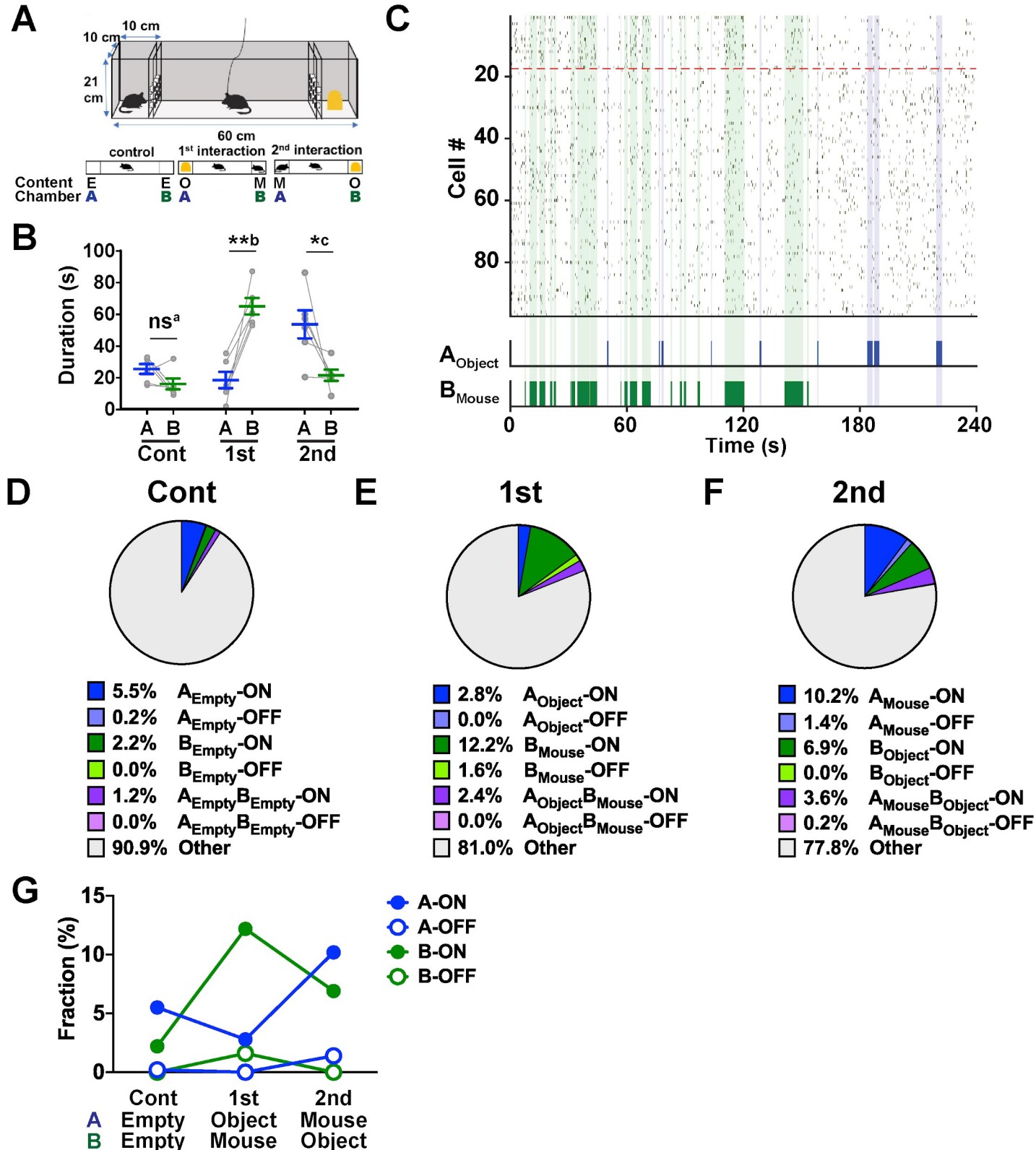

**Fig 3. Behavior and AI neuron activity during LC experiments.** (A) Experimental scheme. Mice were subjected to three 4-min sessions in the LC (top) in the following order (bottom, from left to right): control session in which chambers A and B were empty (E), the first social interaction session in which chamber A and chamber B contained a novel object (O) and a stranger mouse (M), respectively, and the second social interaction session in which chamber A and chamber B contained

a stranger mouse and a novel object, respectively. (B) Total duration of time spent investigating chamber A and chamber B during control ("Cont"), first interaction ("1st"), and second interaction ("2nd") sessions. ns[a], $P = 0.12$, $t_{(5)} = 1.89$; **b, $P = 0.0022$, $t_{(5)} = 5.75$; *c, $P = 0.036$, $t_{(5)} = 2.86$, paired $t$ test, $n = 6$ mice (S1 Data, sheet Fig 3B). (C) A raster plot showing $Ca^{2+}$ events of a population of AI neurons ($n = 99$ cells) imaged in a single experiment during the first interaction session. $B_{Mouse}$-ON cells are sorted above the red dashed lines. The epochs of nose poking to chamber A with a novel object ($A_{Object}$) and chamber B with a stranger mouse ($B_{Mouse}$) are shown in the bottom panel and indicated by blue and green shades, respectively. (D–F) Pie charts showing the fractions of each category of cells during the control session (D), the first interaction session (E), and the second interaction session (F). $n = 580$ cells from 6 mice. (G) Changes in the fractions of A-ON, A-OFF, B-ON, and B-OFF cells across sessions (S1 Data, sheet Fig 3G). The content of each chamber is shown at the bottom. AI, agranular insular cortex; LC, linear chamber; ns, not significant.

limited tactile social cues to the subject mice while preventing full physical contact of the subject mice with the target.

Subject mice spent a comparable amount of time nose poking to empty chambers A and B during control experiments (Fig 3B). During the first interaction session, however, the amount of time nose poking to chamber B, which housed a stranger mouse, substantially increased and was much more than the nose poking to chamber A, which contained a novel object (Fig 3B). During the second interaction session, the subject mice again spent significantly more time nose poking to chamber A with a stranger mouse than to chamber B with a novel object, although the difference in time spent investigating the 2 chambers appeared less prominent than during the first session. The subject mice spent 82.9% ± 4.9% and 84.5% ± 4.5% of total time immobile in the first and second interaction sessions, respectively (mean ± SD, $n = 6$ mice). In addition, mice were almost completely stationary while they were nose poking to each of the test chambers (93.0% ± 4.5% and 97.5% ± 1.1%, chamber A and chamber B during the first interaction session, respectively; 96.7% ± 3.8% and 95.1% ± 3.4%, chamber A and chamber B in the second interaction sessions, respectively; mean ± SD, $n = 6$ mice). Overall, these results demonstrate that a subject mouse investigates the chamber that contains a stranger mouse more than that of the novel object, irrespective of the chamber's location.

We then sought neuronal activity correlated or anticorrelated with the investigation of chamber A and/or B (Fig 3C). Of 580 cells from 6 mice, 5.5% (32 cells) and 2.2% (13 cells) of total cells showed activity significantly correlated with the period of nose poking to empty chamber A and chamber B, respectively, during control experiments (Fig 3D). These 2 types of cells showed higher calcium event rates during investigation of chambers that their activity was correlated with compared to the other chamber (S3A Fig). We therefore categorized these cell types as "$A_{Empty}$-ON cells" and "$B_{Empty}$-ON cells," in which the labels represents the chamber they responded to, followed by the content of the chamber in the subscript and the type of response ("ON" for activation and "OFF" for suppression). Similarly, 1.2% (7 cells) of total cells exhibited activity significantly correlated with investigation of chamber A as well as chamber B ($A_{Empty}B_{Empty}$-ON cells; Fig 3D and S3A Fig). We identified no or a very low fraction of cells that showed activity significantly anticorrelated with investigation of chamber A ($A_{Empty}$-OFF cells) or chamber B ($B_{Empty}$-OFF cells) (Fig 3D).

In the first interaction session, the fractions of cells that exhibited activity significantly correlated ($B_{Mouse}$-ON cells) and anticorrelated ($B_{Mouse}$-OFF cells) with social investigation toward chamber B increased to 12.2% (71 cells) and 1.6% (9 cells) of total cells, whereas those that showed activity correlated and anticorrelated with nonsocial investigation toward chamber A remained low ($A_{Object}$-ON cells, 2.8% [16 cells]; $A_{Object}$-OFF cells, 0.0% [0 cells]; Fig 3E–3G and S3B Fig). During the second interaction session, in which the chambers that contained a stranger mouse and a novel object were exchanged, the fractions of cells whose activity was significantly correlated ($A_{Mouse}$-ON cells) and anticorrelated ($A_{Mouse}$-OFF cells) with nose poking to chamber A increased to 10.2% (59 cells) and 1.4% (8 cells) of total cells, whereas those correlated and anticorrelated with nose poking to chamber B decreased to 6.9% (40 cells; $B_{Object}$-ON cells) and 0.0% (0 cells; $B_{Object}$-OFF cells) of total cells (Fig 3F and 3G and S3C

Fig). Similar changes in the corresponding cell fractions were observed when a female stranger mouse was used instead of a male stranger mouse (S4 Fig). These results indicate that social stimuli activate a larger fraction of AI neurons compared to nonsocial stimuli. Moreover, this increase occurs independent of the locations at which the subjects investigate the social targets and accompanies parallel increases of cells that reduce their activity during social investigation.

We further examined whether the preference of activity of the cells of each category persisted across sessions. More specifically, we focused on the cells of the following categories (Fig 4A): (1) cells that were consistently activated or suppressed during investigation of chambers with stranger mice or with novel objects (here termed "Social cells" and "Object cells" for convenience), (2) cells that were activated or suppressed during investigation of chamber A and chamber B only when they were not empty ("Conditional chamber A cells" and "Conditional chamber B cells"), and (3) cells that were activated or suppressed during investigation of chamber A and chamber B regardless of whether the chambers were empty or not ("Chamber A cells" and "Chamber B cells"). We found that 2.8% (16 cells) of total cells (i.e., 19.7% of Social-ON cells and 22.2% of Social-OFF cells in the first interaction sessions) were Social cells, and 0.9% (5 cells) of total cells were Conditional chamber B cells (Fig 4B and 4C and S5 Fig). The observed number of consistent Social-ON cells was about twice the number expected from cell fractions in each individual session (observed, $2.33 \pm 1.03$ cells/FOV; expected, $1.08 \pm 0.47$ cells/FOV, mean $\pm$ SD, $n = 6$ mice; Fig 4D). Besides these cells, we found no or a very small number of cells of other specific categories (Fig 4B, see the legend). These results indicate that neurons that consistently encode the investigation of nonsocial stimuli or spatial information per se are very rare in the AI.

Finally, we examined whether insular social cells identified in the 2 different behavioral tasks formed a shared neuronal subset. Among 71 $B_{Mouse}$-ON cells identified during the first interaction session of the LC experiment, 16 $B_{Mouse}$-ON cells (22.5%, found in 6 out of 6 subject mice) were also Social-ON cells in HC experiments, while none of 9 $B_{Mouse}$-OFF cells were Social-OFF cells in HC experiments (S6 Fig). However, the extent of overlap of these two ON cell ensembles did not differ significantly from chance level (observed, $0.158 \pm 0.099$ versus expected, $0.123 \pm 0.035$, mean $\pm$ SD, $P = 0.44$, $t_{(5)} = 0.85$, paired $t$ test, $n = 6$ mice; S6C Fig), suggesting that, unlike the more consistent representations of social exploration within the same experiments (Fig 4D), those across different tasks, contexts, and/or days are rather independent from each other.

## Discussion

Social behavior requires multiple steps of information processing whereby multimodal sensory inputs are transformed into appropriate behavioral outputs via social decision-making. The IC is anatomically situated to integrate multisensory social cues to influence the activity of the evolutionarily conserved "social decision-making network," which consists of the core nodes of the "social behavior network" and the mesolimbic reward system [32, 35, 36]. Our study directly observed the activity of AI neurons during social interaction at a single cell level and thus provides new insight into the cellular basis of the social function of the AI. We identified a larger fraction of Social-ON cells and a smaller fraction of Social-OFF cells that exhibit opposing activity during social exploration. Social-ON cells included those that represented social investigation independent of location and consisted of multiple subsets, each of which were preferentially active during exploration under a particular behavioral state or with a particular target of physical contact. These findings imply that neuronal ensembles in the AI encode the ongoing status of social exploration while an animal interacts with unfamiliar

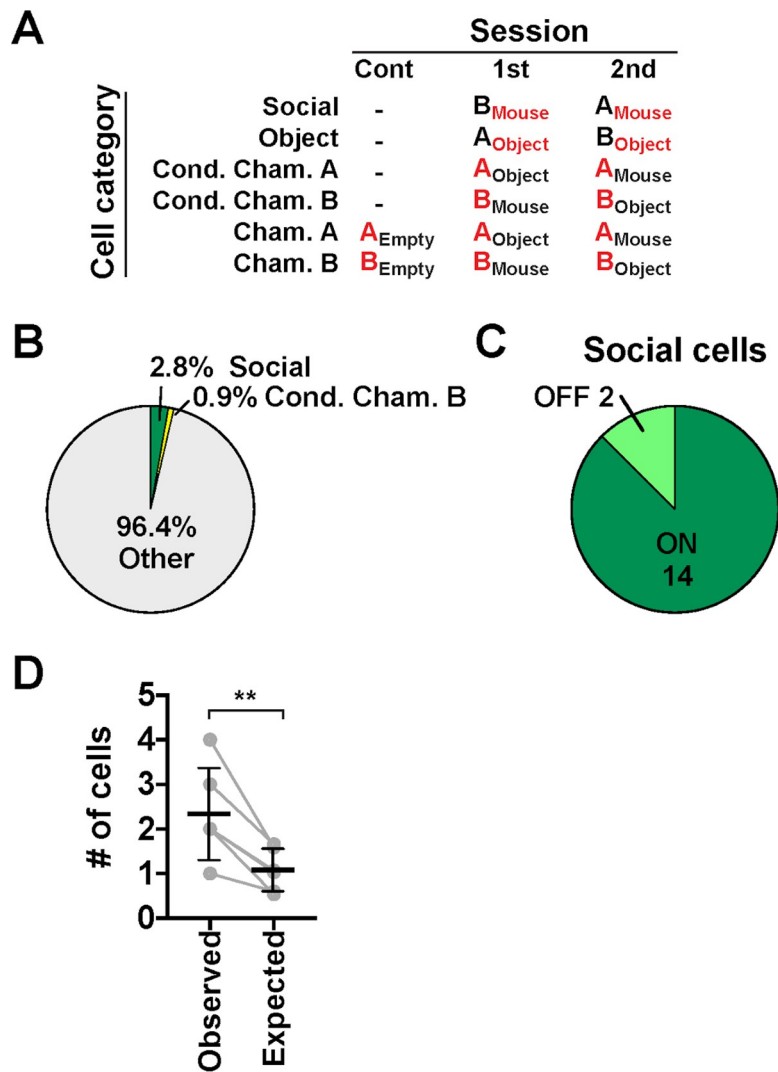

**Fig 4. Social cells and other cell categories in LC experiments.** (A) Definition of each category of cells and their preference for stimulus–chamber relationship. "A" and "B" represent chamber A and chamber B, respectively. Their contents are indicated by the subscript to represent stimulus–chamber relationship. The features each cell category preferentially responds to are shown in red. (B) A pie chart that shows the fractions of each cell category within the total cell population ($n$ = 580 cells from 6 mice). "Other" includes very small fractions of chamber A cells (0.3%), conditional chamber A cells (0.2%), and object cells (0.2%). (C) A pie chart showing the fractions of "ON" cells and "OFF" cells within total social cells. Numbers represent absolute cell numbers. (D) The number of consistent Social-ON cells observed ("Observed") and that expected from the fraction of cells in each session ("Expected") (mean ± SD; $P$ = 0.0063, observed versus expected, $t_{(5)}$ = 4.52, paired $t$ test, $n$ = 6 mice; S1 Data, sheet Fig 4D). LC, linear chamber.

conspecifics. However, insular neurons are also shown to be able to encode internal states independent of behavior [37]. It therefore remains to be investigated whether social cells in the AI encode interaction with "other" and/or changes in the state of "self" associated with social exploration.

Recent rodent studies have demonstrated the role of the anterior IC in relapse to drug-seeking and alcohol self-administration [38, 39] and that of the posterior IC in social emotional behavior and aversive state processing [18–20]. In addition to these functional characterizations, contrasting gradient distribution of projections to the core of the nucleus accumbens and the central amygdala [18], distinct taste and valence coding in the gustatory IC [22], and

intensive functional and anatomical intra-insular connectivity [18, 19] suggest that anterior and posterior ICs serve distinct yet interrelated functions. Our identification of social exploration-tuned neurons in the AI is in accordance with activity-regulated gene expression in the AI of male mice after brief social encounter [40]. Although optogenetic and chemogenetic manipulations of posterior IC activity did not appear to affect sociality per se, altered social exploration of male rats by pharmacological blockade of NR2B subunit-containing N-methyl-D-aspartate-type glutamate receptors, in the AI, suggests that the activity of social cells in the AI may influence social behavioral output [41]. Thus, future studies such as identification and manipulation of the projection targets of Social-ON and Social-OFF cells and comparison of their activity between anterior and posterior ICs in a socioemotional task will greatly advance our understanding of their significance and circuit mechanisms. Intriguingly, similar ON and OFF neural ensembles that code social exploration have been reported in the prelimbic area of the mPFC [12], raising the hypothesis that this mode of encoding is common across cortical areas.

The higher fraction of social cells observed in HC experiments compared with LC experiments may reflect the difference in direct contact to social targets, which not only includes tactile cues but is also known to activate the accessory olfactory bulb when combined with pheromonal cues [42]. Social facial touch with conspecifics elicits stronger responses than object touch in the rodent barrel cortex [43]. Furthermore, cortical sensory processing is strongly influenced by locomotion [44–46]. Unique combinations of multimodal social cues detected at different targets of physical contact and locomotion state therefore likely drive a subset of Social-ON cells preferentially over others. In LC experiments, social stimulus markedly increased the number of AI neurons that changed their activity upon nose poking to that chamber. Although neurons consistently encoding the chamber location were hardly found, presentation of a stranger mouse increased the neurons that were responsive to this chamber. These findings highlight an important feature of AI neurons in that they encode exploration of salient stimuli rather than spatial information itself. Longer exploration of unfamiliar conspecifics suggests that they represent more salient and attractive stimuli than inanimate objects. In humans, affective touch activates the posterior IC and elicits increases in functional connectivity between the posterior IC and anterior IC, a key node of the "salience network" [47, 48]. This posterior-to-anterior progression is proposed to reflect the sequential integration of internal emotional and bodily state signals with environmental, motivational, and cognitive conditions to form the percept of self-awareness that guides appropriate behavioral responses [14, 16]. This view accords well with the notion that the IC acts at the interface of "social" and "emotional" modules of the "social decision-making network" in the rodent brain [19]. Consistent with these ideas, our findings imply that social cells in the AI encode not only the online status of social interaction at an individual cell level but also the social salience of the target that the subject is interacting with at a population level. Finally, the structure and function of the IC are known to be altered in various brain disorders, including schizophrenia and ASD. Our study thus provides a cellular basis of insular social function on which future research using mouse models of neuropsychiatric and neurodevelopmental disorders will be grounded [49, 50].

## Materials and methods

### Ethics statement

All experimental procedures were conducted in accordance with institutional guidelines and protocols approved by the RIKEN Animal Experiments Committee. Our protocol (W2019-2-042) adhered to the Fundamental Guidelines for Proper Conduct of Animal Experiment and

Related Activities in Academic Research Institutions under the jurisdiction of the Ministry of Education, Culture, Sports, Science and Technology.

## Mice

Sexually inexperienced adult male C57BL/6J mice (8–12 wk old, Japan SLC, Hamamatsu, Japan) were used. Mice were maintained on a 12 h-12 h light-dark cycle with ad libitum access to food and water. All surgical manipulations and behavioral tests were conducted during the dark phase of the cycle.

## Surgery

Each mouse underwent 2 separate surgical procedures. In the first surgery, the mouse was subjected to microinjection of an adeno-associated viral (AAV) vector that drove neuronal expression of a genetically encoded calcium indicator and insertion of a gradient refractive index (GRIN) lens into the AI. In the second surgery, a baseplate for the miniaturized head-mounted microscope (nVista, Inscopix, Palo Alto, CA) was affixed onto the skull.

The AAV vector injection and GRIN lens insertion were conducted on the same day. Mice anaesthetized with 2% isoflurane in oxygen at a flow rate of 0.5 L/min were placed in a stereotactic setup (Kopf Instruments, Tujunga, CA). Body temperature was maintained at 37°C with a heating pad (40-90-2-05, FHC, Bowdoin, ME). Ophthalmic ointment (Sato Pharmaceuticals, Tokyo, Japan) was applied to the eyes to prevent drying. A piece of scalp was removed, and a small craniotomy was performed on the skull above the right AI using a high-speed rotary micro drill (OmniDrill 35, World Precision Instruments, Sarasota, FL). A 35-gauge needle attached to a microsyringe (World Precision Instruments) was targeted to the right AI (1.94 mm anterior, 2.2 mm lateral, and 3.45 mm in depth relative to bregma), and 500 nl of virus solution (AAV5-CaMKII-GCaMP6f-WPRE-SV40, $1.0 \times 10^{14}$ GC/mL, Penn Vector Core) was microinjected at a rate of 100 nl/min. The needle was left undisturbed for an additional 10 min to avoid backflow. The needle was then slowly withdrawn, and the injected surface was washed with saline. Following the microinjection, a GRIN lens (0.5 mm diameter, 4.0 mm length, GLP-0540, Inscopix, Palo Alto, CA) attached to a lens implant kit (ProView, Inscopix, Palo Alto, CA) was slowly inserted stereotaxically to a position slightly dorsal to the AAV injection site (1.94 mm anterior, 2.2 mm lateral, and 3.25 mm in depth relative to bregma). The lens was then affixed to the skull with dental cement, and the top of the lens was covered with a silicon mold (Kwik-Cast, World Precision Instruments, Sarasota, FL). Mice were singly housed after fully recovering from the surgery.

Four weeks after the GRIN lens implantation, mice were anaesthetized again with isoflurane, and the silicon mold over the tip of the GRIN lens was carefully removed. The baseplate (Inscopix) attached to a miniaturized head-mounted microscope was positioned above the implanted lens using an adjustable gripper (Inscopix). The microscope and baseplate were then lowered toward the top of the lens until the FOV was in focus. After confirmation of GCaMP fluorescence signals, the baseplate was affixed to the skull using dental cement. The baseplate was covered with a baseplate cover (Inscopix) after the microscope was detached from the baseplate.

## Social behavior tests

We used 2 different behavioral assays, the HC experiment and the LC experiment, for social behavior tests. Subject mice were singly housed in their HCs for at least 4 wk before the exposure to social stimuli. The behavioral schedule consisted of at least 2 d of habituation and a day of testing for HC experiments, followed by at least 2 d of habituations and a day of testing for

LC experiments. A total of 9 mice were used for these experiments. Data from all 9 mice were analyzed in HC experiments, and data from 3 mice were excluded in the analysis of LC experiments because the image quality was low. The behavior was recorded using a video camera (logicool) throughout the test in both paradigms. Different stranger mice were used in HC experiments and LC experiments.

In the HC experiment, a subject mouse with a microscope attached to its head underwent habituation sessions in which the mouse was allowed to move freely in its HC (30 cm long, 18 cm wide, and 12 cm high) for at least 20 min/d. On the day of imaging, a set of tests that consisted of a control session and an interaction session were performed as follows (Fig 1A). First, a novel miniature object (nonsocial target) and a microscope-attached subject mouse were placed together in its HC, and the mouse was allowed to move freely in this environment for 4 min. Immediately after this control session, the object was replaced with a stranger mouse (social target), and the subject mouse was allowed to freely explore for an additional 4 min. The subject mouse remained in the test environment during the brief interval between the 2 sessions.

In the LC experiment, an acrylic LC that consisted of a center chamber (40 cm long, 10 cm wide, and 21 cm high) flanked by 2 test chambers (chambers A and B; 10 cm long, 10 cm wide, and 21 cm high each) was used (Fig 3A) [34]. The bottom 10 cm of the walls that divided the center chamber and the test chambers had 1-cm-diameter holes drilled to allow subject mice to interact with the targets by nose poking. Each test chamber contained a male stranger mouse (social stimulus), a miniature object (nonsocial stimulus), or nothing (no stimulus). In some experiments ($n = 2$ additional mice), a female stranger mouse was used instead of a male stranger mouse. At least 2 d prior to imaging experiments, a subject mouse with a microscope attached to its head was allowed to move freely in the center chamber during a 20-min habituation session per day for 2 d. The test consisted of 3 consecutive sessions (Fig 3A). In the first session, a subject mouse placed in the center chamber was allowed to move freely for 4 min while no stimuli were presented in the 2 test chambers (control session). In the following session, a novel miniature object and a stranger mouse were placed in chamber A and chamber B, respectively, and the subject mouse was allowed to freely explore in the center chamber for an additional 4 min (first interaction session). In the last session, the positions of the stranger mouse and the object were swapped, and the subject mouse was allowed to explore freely for another 4 min (second interaction session).

## Behavioral data analysis

The behavior of subject mice was manually classified by visual inspection of the videos after the experiments. In HC experiments, we defined the periods of social interaction as those during which the subject mouse actively explored the social target by nasal contact, and these periods were further subclassified into the periods of contact to the nose, body, and anogenital area of the stranger mouse. The remaining periods were categorized as nonsocial periods. Within the nonsocial periods, the periods during which the subject mouse touched the wall of the HC and those during which the subject mouse was passively touched by the stranger mouse were documented. For analysis of behavioral states, the periods during which the subject mouse changed its body and hindlimb positions were defined as moving periods, and the remaining periods of immobility were classified as stationary periods. In LC experiments, we defined the periods of investigation of the test chamber as nose poking by the subject mouse to the wall that divided the center chamber and the test chamber.

## Ca$^{2+}$ imaging of AI neurons in freely moving mice

Activity of GCaMP6f-labeled AI neurons during social behavior tests was imaged using a miniaturized head-mounted microscope and GRIN lens-mediated microendoscopy (Fig 2A).

Before the experiments, mice were lightly anesthetized with 2% isoflurane, and the baseplate cover was removed from the baseplate, to which a miniaturized head-mounted microscope was subsequently attached. Mice were then recovered from anesthesia for at least 20 min before beginning the experiments.

Ca$^{2+}$ imaging videos were recorded during social behavior tests using nVista acquisition software (version 1.2.0; Inscopix, Palo Alto, CA) with a resolution of $1,440 \times 1,080$ pixels at a rate of 15 frames/s. The LED power of the microscope was set between 30% and 50% (0.36–0.6 mW). Ca$^{2+}$ transients were observed in many neurons within the FOV while mice moved freely in their environments (Fig 2B, $81.9 \pm 44.6$ cells/FOV, mean $\pm$ SD, $n = 9$ mice), and post hoc confirmation of the lens positions verified that GRIN lenses were successfully targeted to the AI in all 9 cases (Fig 2C).

## Imaging data analysis

Post-acquisition processing of Ca$^{2+}$ imaging videos was performed using the Inscopix Data Processing Software (IDPS; version 1.3; Inscopix). Videos were spatially downsampled by a factor of 4. Motion correction was performed by shifting each frame to a single reference frame so that high contrast features within each frame were aligned to the corresponding features in the reference frame. Normalized fluorescence changes were then visualized in a $\Delta F/F_0$ video, in which a minimum z-projection image of the entire movie ($F_0$) was subtracted from each frame (F), and the resultant $F–F_0$ movie was normalized to $F_0$. Cells in the $\Delta F/F_0$ video were then identified using an extended constrained non-negative matrix factorization algorithm for microendoscopic data (CNMF-E; parameters, min_corr = 0.8, min_pnr = 5–10, gSig = 3, gSiz = 30) [51, 52], followed by human visual verification against the $\Delta F/F_0$ video, in which candidate cells that did not show appropriate cell morphology and fluorescence intensity changes were removed manually from further analysis. Cells in LC experiments were identified from single concatenated movies that spanned the control, the first interaction, and the second interaction sessions, and thus all the cells were completely registered across these sessions. Ca$^{2+}$ transient events were detected using a Ca$^{2+}$ event detection algorithm in IDPS (parameters, event smallest decay time = 0.20 s, event threshold factor = 5 median absolute deviation) by finding large peaks of fluorescence changes with fast rise times and exponential decay.

Neurons that exhibited activity correlated and anticorrelated with social behavior were searched by statistical testing of the similarity index calculated between the activity vector A and the behavior vector B [12, 53, 54]. The activity vector A was a binary vector that represented the occurrence of neuronal activity event by 1, and the behavior vector B was a binary vector that represented the period of social interaction by 1. Identification of neurons whose activity was correlated with other types of behavior (i.e., movement, immobility, etc.) was conducted similarly by using the behavior vector that represented the occurrence of the behavior of interest by 1. The similarity index, also known as cosine similarity, was defined as normalized inner product of these two vectors by the following equation:

$$similarity = \frac{\sum_{i=1}^{n} A_i B_i}{\sqrt{\sum_{i=1}^{n} A_i^2} \sqrt{\sum_{i=1}^{n} B_i^2}}$$

The similarity index ranges from 0 to 1, in which a value of 1 indicates that the two vectors are identical and a value of 0 indicates that they are orthogonal. We then compared this value to a distribution of the similarity index expected by chance, which was obtained by calculation using 3,600 randomly shuffled activity vectors derived from the same cell. The statistical significance level ($P = 0.05$) was corrected for multiple comparisons by the Bonferroni method, in

which the *P* value was divided by the number of tests performed (*n* = 11 tests for HC experiments: social interaction, nose, body, anus, passive, wall, stationary, movement, social-stationary, social-movement, and no interaction; *n* = 4 tests for LC experiments: chamber A-ON, chamber A-OFF, chamber B-ON, and chamber B-OFF). In HC experiments, a cell's activity was considered to be significantly correlated with the behavior if its similarity index in the real data was greater than the 99.55% quantile. The population of Social-ON cells included cells whose activity was significantly correlated with social interaction, nose, body, anus, social-stationary, and/or social-movement. Social-OFF cells were defined as those whose similarity index was lower than the 0.45% quantile of the distribution predicted by chance. In LC experiments, a cell's activity was considered to be significantly correlated and anticorrelated with the investigation of a chamber if its similarity index in the real data was greater than the 98.75% quantile and lower than the 1.25% quantile of the distribution predicted by chance, respectively. Chamber AB-ON cells and chamber AB-OFF cells were defined as those whose activity was significantly correlated and anticorrelated with the investigation of both chambers, respectively.

Ca$^{2+}$ event rate of each neuron during a particular behavior was calculated as the number of Ca$^{2+}$ transient events during the periods of the behavior of interest divided by the total length of the periods. To quantitatively estimate the preference of activity of each neuron for social interaction, we calculated the social preference index as (S − N) ÷ (S + N), where S and N are Ca$^{2+}$ event rates of the neuron during social interaction and nonsocial periods, respectively. The social preference index ranges from −1 to 1, where a positive value indicates a preference for social interaction and a negative value indicates a preference toward nonsocial periods.

Calculation of NNDs between Social-ON cells was conducted by averaging NNDs across all Social-ON cells within the map. To test whether the anatomical distribution of Social-ON cells was random, the same numbers of labels of Social-ON cells as the real data were randomly shuffled 1,000 times, and an average of NNDs calculated from the shuffled dataset was compared to the values obtained from the real data.

Identification of Social cells that were common between HC experiments and LC experiments were conducted by tracking individual cells in the 2 social cell maps using the longitudinal registration function in IDPS (parameter, minimum correlation = 0.7), which registers and matches cells from different sessions by finding the pair of cells that maximize the spatial correlation of their images [55, 56]. Under this condition, 70.5% (432 of 613 cells from 6 mice) of cells identified in HC experiments and 74.5% (432 of 580 cells from 6 mice) of cells in LC experiments were registered with their counterpart cells that were most likely the same cell in the other map. The extent of overlap between the 2 ensembles A and B was estimated by calculating the Sørensen–Dice coefficient as follows (S6C Fig) [12]:

$$Overlap\,(A, B) = \frac{2|A \cap B|}{|A| + |B|}$$

This index was then compared to that expected by chance, which was calculated by $2N_AN_B \div (N[N_A + N_B])$, where $N_A$ and $N_B$ are the number of cells in each ensemble and $N$ is the number of cells in the union of total cells identified in HC experiments and LC experiments.

The number of consistent Social-ON cells in each mouse in LC experiments was compared to that expected by chance (Fig 4D) [12], which was calculated by $N_AN_BN_C/N^2$, where $N_A$, $N_B$, and $N_c$ are the number of cells in each ensemble and $N$ is the number of total cells identified.

## Histology

Mice deeply anesthetized with isoflurane were transcardially perfused with phosphate-buffered saline (PBS) followed by 4% paraformaldehyde (PFA) in PBS. Brains were dissected out and post-fixed in 4% PFA overnight; 50-μm-thick coronal sections were cut on a vibratome (Leica VT1200S). Fluorescence images were acquired using a Keyence BZ-9000 epifluorescence microscope (Keyence, Osaka, Japan).

## Statistics

Data are represented as mean ± SEM unless otherwise stated. Statistical tests were performed using GraphPad Prism version 8 (GraphPad Software, La Jolla, CA). All two-group statistical tests were two-sided, and statistical significance was defined as $P < 0.05$. Exact $P$ values are shown unless $P < 0.0001$.

## Supporting information

**S1 Fig. Modulation of social cell activity by social interaction.** (A) Box plots of $Ca^{2+}$ event rates of Social-ON ("Soc-ON," $n = 168$ cells), Social-OFF ("Soc-OFF," $n = 10$ cells), and nonsocial cells ("Non-Soc," $n = 559$ cells) during nonsocial periods (NS) and social interaction periods (S). Whiskers represent 10–90 percentile, and red dots represent outliers. ***a, $P < 0.0001$, $W_{(168)} = 14,196$, $n = 168$ cells; **b, $P = 0.0020$, $W_{(10)} = -55$, $n = 10$ cells; Wilcoxon matched-pairs sign rank test; ***c, $P < 0.0001$, $U_{(168, 559)} = 30,636$; $n = 168$ and 559 cells; ***d, $P < 0.0001$, $U_{(10, 559)} = 673$; $n = 10$ and 559 cells; Mann-Whitney test (S1 Data, sheet S1A Fig). (B) Distribution of social preference indices of individual neurons. The fractions of Social-ON cells, Social-OFF cells, and nonsocial cells are shown in red, blue, and gray in stacked bars (S1 Data, sheet S1B Fig).
(TIF)

**S2 Fig. $Ca^{2+}$ event rates of each cell type during behavior in HC experiments.** (A) GCaMP6f fluorescence change of a Social-stationary cell (top) and a Social-movement cell (bottom). (B) $Ca^{2+}$ event rates of Social-stationary cells ("Soc-stat"), Social-movement cells ("Soc-move"), and other Social-ON cells ("other") during social interaction with ("S-Mo") and without ("S-St") movement. ***a, $P < 0.0001$, $W_{(101)} = 5,075$; ***b, $P = 0.0005$, $W_{(12)} = -78$; Wilcoxon matched-pairs sign rank test (S1 Data, sheet S2B Fig). (C) GCaMP6f fluorescence change of nose (top), body (middle), and anus (bottom) subtypes of Social-ON cells. (D) $Ca^{2+}$ event rates of nose, body, anus, and other subtypes of Social-ON cells during social interaction with contact with nose ("N"), body ("B"), and anus ("A"). Since the fraction of time spent contacting anus was low, only event rates during contact with nose and body are shown for nose, body, and other cell subtypes. ***a, $P < 0.0001$, $W_{(60)} = -1,830$; ***b, $P < 0.0001$, $W_{(34)} = 595$; Wilcoxon matched-pairs sign rank test.; **c, $P = 0.0012$ versus N; *d, $P = 0.014$ versus B; Friedman test with Dunn's multiple comparisons test (S1 Data, sheet S2D Fig).
(TIF)

**S3 Fig. $Ca^{2+}$ event rates of each cell type in LC experiments.** (A) Box plots of $Ca^{2+}$ event rates of Chamber A-ON cells ($A_{Empty}$-ON, $n = 32$ cells), Chamber B-ON cells ($B_{Empty}$-ON, $n = 13$ cells), Chamber AB-ON cells ($A_{Empty}B_{Empty}$-ON, $n = 7$ cells), and other cells (Other, $n = 527$ cells) during the periods when the subject mice investigated Chamber A ("A"), Chamber B ("B"), or otherwise ("C") in control sessions. Whiskers represent 10–90 percentile, and red dots represent outliers. Cell categories whose fractions are larger than 1% are shown. ***a, $P < 0.0001$; *b, $P = 0.018$; *c, $P = 0.023$; **d, $P = 0.0099$; ns^e, $P > 0.99$; Friedman test with Dunn's multiple comparisons test (S1 Data, sheet S3A Fig). (B) Box plots of $Ca^{2+}$ event rates of

Chamber A-ON cells ($A_{Object}$-ON, $n$ = 16 cells), Chamber B-ON cells ($B_{Mouse}$-ON, $n$ = 71 cells), Chamber B-OFF cells ($B_{Mouse}$-OFF, $n$ = 9 cells), Chamber AB-ON cells ($A_{Object}B_{Mouse}$-ON, $n$ = 14 cells), and other cells ($n$ = 470 cells) in the first interaction session. **a, $P$ = 0.0044; ***b, $P$ < 0.0001; **c, $P$ = 0.0096; **d, $P$ = 0.001; ***e, $P$ < 0.0001; Friedman test with Dunn's multiple comparisons test (S1 Data, sheet S3B Fig). (C) Box plots of $Ca^{2+}$ event rates of Chamber A-ON cells ($A_{Mouse}$-ON, $n$ = 59 cells), Chamber A-OFF cells ($A_{Mouse}$-OFF, $n$ = 8 cells), Chamber B-ON cells ($B_{Object}$-ON, $n$ = 40 cells), Chamber AB-ON cells ($A_{Mouse}B_{Object}$-ON, $n$ = 21 cells), and other cells ($n$ = 451 cells) in the second interaction sessions. ***a, $P$ < 0.0001; ***b, $P$ = 0.0009; ***c, $P$ < 0.0001; **d, $P$ = 0.0021; ***e, $P$ < 0.0001; Friedman test with Dunn's multiple comparisons test (S1 Data, sheet S3C Fig).
(TIF)

**S4 Fig. AI neuron activity during exploration of a chamber with a female stranger.** (A) A raster plot showing $Ca^{2+}$ events of a population of AI neurons ($n$ = 61 cells) imaged in a single experiment during the first interaction session with a female stranger. $B_{Female}$-ON cells are sorted above the red dashed lines. The epochs of nose poking to chamber A with a novel object ($A_{Object}$) and chamber B with a female stranger ($B_{Female}$) are shown in the bottom panel and indicated by blue and green shades, respectively. (B). Changes in the fractions of A-ON, A-OFF, B-ON, and B-OFF cells across sessions ($n$ = 105 cells from 2 mice; S1 Data, sheet S4B Fig). The content of each chamber is shown at the bottom.
(TIF)

**S5 Fig. Social cells consistent across multiple LC sessions.** (A). GCaMP6f fluorescence change of a Social-ON cell during control (top, "Cont"), first interaction (middle, "1st"), and second interaction sessions (bottom, "2nd") of LC experiments. (B) GCaMP6f fluorescence change of a Social-OFF cell during control (top), first interaction (middle), and second interaction sessions (bottom) of LC experiments.
(TIF)

**S6 Fig. Social cells common across different tests in the AI.** (A) Example social cell maps of LC experiments (left) and HC experiments (right) imaged in the same animal. $B_{Mouse}$-ON cells and $B_{Mouse}$-OFF cells in the first interaction session of LC experiments and Social-ON cells and Social-OFF cells in HC experiments are shown in red and blue, respectively. The ON cells common to the two paradigms (common Social-ON cells) are indicated by asterisks. (B) GCaMP6f fluorescence change of a common Social-ON cell during the first interaction session of LC experiment (top) and HC experiment (bottom). (C) Overlap of the ON cell ensembles observed between HC experiments and LC experiments ("Observed") and that expected by chance ("Expected") (paired $t$ test, $n$ = 6 mice; S1 Data, sheet S6C Fig).
(TIF)

**S1 Movie. Microendoscopic calcium imaging of AI neurons in freely moving mice.** This movie represents raw GCaMP6f fluorescence. Time stamps are indicated in the bottom right corner. The playback speed is 2x faster than the original recording.
(AVI)

**S1 Data. The numerical data for the graphs presented in this paper.**
(XLSX)

## Acknowledgments

We thank Masanori Nomoto, Thomas McHugh, and Taro Toyoizumi for discussion.

## Author Contributions

**Conceptualization:** Masaaki Sato, Toru Takumi.

**Data curation:** Isamu Miura.

**Formal analysis:** Isamu Miura.

**Funding acquisition:** Toru Takumi.

**Investigation:** Isamu Miura.

**Methodology:** Isamu Miura, Eric T. N. Overton, Nobuo Kunori.

**Project administration:** Masaaki Sato, Junichi Nakai, Takakazu Kawamata, Toru Takumi.

**Supervision:** Nobuhiro Nakai, Toru Takumi.

**Validation:** Masaaki Sato, Nobuhiro Nakai, Toru Takumi.

**Writing – original draft:** Isamu Miura, Masaaki Sato.

**Writing – review & editing:** Masaaki Sato, Eric T. N. Overton, Toru Takumi.

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
