## [Editor Report · Decision Letter 0]

14 Nov 2019

Dear Dr Takumi, 

Thank you for submitting your manuscript entitled "Encoding of social exploration by neural ensembles in the insular cortex" for consideration as a Short Report by PLOS Biology.

Your manuscript has now been evaluated by the PLOS Biology editorial staff, as well as by an Academic Editor with relevant expertise, and I am writing to let you know that we would like to send your submission out for external peer review.

Please re-submit your manuscript within two working days, i.e. by Nov 18 2019 11:59PM.

Kind regards,

Gabriel Gasque, Ph.D.,

Senior Editor

PLOS Biology

---

## [Decision Letter · Decision Letter 1]

19 Dec 2019

Dear Dr Takumi,

Thank you very much for submitting your manuscript "Encoding of social exploration by neural ensembles in the insular cortex" for consideration as a Short Report at PLOS Biology. Your manuscript has been evaluated by the PLOS Biology editors, by an Academic Editor with relevant expertise, and by three independent reviewers.

In light of the reviews (below), we will not be able to accept the current version of the manuscript, but we would welcome re-submission of a much-revised version that takes into account the reviewers' comments. We cannot make any decision about publication until we have seen the revised manuscript and your response to the reviewers' comments. Your revised manuscript is also likely to be sent for further evaluation by the reviewers.

We expect to receive your revised manuscript within 2 months. 

**IMPORTANT - SUBMITTING YOUR REVISION**

Your revisions should address the specific points made by each reviewer. As you’ll see from their comments, all three reviewers agree that a solid paper assessing the cellular level function of the insular cortex in social behaviors would be a suitable advance for PLOS Biology. However, reviewer 1 in particular raised a number of fairly serious technical/data analysis concerns that s/he feels limit your ability to draw strong conclusions. Further, both reviewers 1 and 3 question whether the current findings alone provide a sufficient level of advance absent additional types of mechanistic insights. We do, however appreciate that this manuscript was submitted as a Short Report. Therefore, if you are able to satisfy the technical concerns of the reviewers, and sufficiently revise the presentation of the study as highlighted by reviewer 2, we would be willing to consider a revision of this study at PLOS Biology. This means that we will not require further mechanistic insights for publication in our journal, but we would welcome them, nonetheless, if you have additional data already in hand. 

Regardless the nature of your revisions, please keep the number of main figures at a maximum of four. 

Please submit the following files along with your revised manuscript:

*NOTE: In your point by point response to to the reviewers, please provide the full context of each review. Do not selectively quote paragraphs or sentences to reply to. The entire set of reviewer comments should be present in full and each specific point should be responded to individually, point by point.

*Re-submission Checklist*

*Published Peer Review*

*PLOS Data Policy*

*Blot and Gel Data Policy*

Sincerely,

Gabriel Gasque, Ph.D., 

Senior Editor

PLOS Biology

REVIEWS:

Reviewer #1: Miura et al. employed microendoscopic calcium imaging to study the social behavior related neural ensembles in the agranular insular cortex (AI). The authors found that the activity of two neural ensembles, a larger ensemble of Social-ON neurons and a smaller ensemble of Social-OFF neurons, changed in opposite directions during social exploration. More specifically, Social-ON cells encode social investigation independent of location and were preferentially active during exploration. Encoding of social behavior in the AI is an important subject that is relevant to many psychiatric disorders. Therefore, this manuscript has potential significance and novelty to be suitable for publication in PLOS Biology. However, first and foremost, the authors need to revisit their data analysis. It has been shown in the literature that CNMFe algorithm outperforms PCA/ICA algorithm in many regards. The authors should reanalyze their data using the now widely accepted CNMFe algorithm. In the identification of behavior relevant neurons, the authors need to consider correction for multiple comparison, instead of simply use “greater than the 95th percentile and smaller than the 5th percentile of the values obtained form the randomly permuted data”. In its current form, the results in the manuscript are descriptive. The authors didn’t provide sufficient quantitative analyses to support their claims. It is this reviewer’s opinion that this manuscript, in its current form, is not suitable for publication in PLOS Biology. Below are some examples of major concerns and comments:

1. Figure 1 

(a) Figure 1 and FigureS1 should be combined to help reader to understand the results. Figure S1 lacks scale information for both field of view (x-y axis) and calcium traces (amplitude). The signal to noise ratio of the calcium traces seems low from Figure S1B. This reviewer would suggest the authors to add representative calcium images or raw (not heavily processed) calcium video to the manuscript, to convince the readers that some of the cells are not falsely detected by the cell identification algorithm.

(b) Figure 1B indicates that mice did not explore the novel object at all.

(c) Rearrange the panels of current Figure 1 A, B, E, D, C to new A, B, C, D, E in revised Figure1 may help readers to follow the information flow of Figure 1 better.

(d) It is difficult to tell the differences between social cells (above the red line) and non-social cells (below the red line) from Figure 1C. This would make the findings of this manuscript questionable.

2. Figure 2,3

(a) In my personal opinion, it is not necessary to show Figure 2D-E, figure 3E-I. For instance, the Social-ON cells are supposed to have higher activation rate in social time than that in non-social time. Because this is the way that the authors define (identify) them.

(b) Figure 2F needs to add scale bar and change the unit in pixels to actual anatomical scale.

(c) After removing repetitive data from these two figures, Figure2-3 only provide some basic descriptive observations (i.e., number of different type of cells). More mechanistic insights regarding AI cells in coding social behavior need to be provided. 

4. Figure 4. Through this figure together with Figure S4, the authors were trying to find the consistent social cells. Unfortunately, the authors fail to provide any quantitative analysis to support their claim. Moreover, the cell numbers are too low to differentiate from randomness. There is no information in the manuscript detailing how the authors registered the cells in two different experiments in the manuscript, related to Figure S4A in which the two neural maps look quite different, raising concerns about cell registration process across different imaging sessions.

5. Figure S2BDF: no y-axis label

Reviewer #2: This manuscript reports on an experiment in which the activity of mouse agranula insular cortex CamKII neurons was quantified during a social behavior task using in vivo GCAmp and miniature endoscopes (miniscope). This is an understudied area of the brain that very likely plays important roles in social and emotional behavior and the use of miniscopes in this region is a first, to my knowledge in a social behavior setting. Thus the novelty and relevance of this report is potentially very high. It seems that the experimental procedures and analysis of miniscope data are consistent with the standard in the field and the new data can be useful to shed light on the contribution of insula to social behavior/cognition; however the manuscript discussion makes many speculations that are well beyond the supporting data, and the key conclusion, that insula provides a sort of online status of social interaction (which seems to be supported by data) is not well framed in the discussion which diminishes enthusiasm for the paper (see below). 

The authors must also consider reducing the number of cell groups discussed. Several paragraphs of results name the number of cells that fall into different categories, accounting for only a very tiny fraction of observed neurons (i.e. 0.8% of cells were anticorrelated with a specific chamber in the apparatus). I found the presentation of results made it very difficult to separate the meaningful and more abundant types of cells from these very obscure ones. Some cells are likely to correlate with certain events merely by chance, how did the authors decide which populations should be included for analysis or represent statistically meaningful groups of cells? I strongly suggest applying a threshold or conducting some probability based analysis and then eliminate or move the presentation of the infrequent cell types (e.g. occurring in fewer than 3-4% of neurons) to a supplemental file or a figure only. The number of different and idiosyncratic labels for the types of cells makes the results section almost impossible to decipher. A minor point related to the presentation, the authors used just one letter to denote many aspects of their design, such as E=empty, O=object. While most of this is acceptable, with so many variations in the test, looking at the figures would be far more intuitive if whole words were used as labels rather than letters only. The consequence of these two points is evident in Figure 3 panels D, F H…it is impossible to make sense of these charts without adding copious additional notes to the panel, and all of this is probably wasted energy because many of these cell categories are simply not enough neurons to discuss in this brief report.

The discussion suggests the Kim et al 2015 paper was the basis for investigating the insula, however, there are many other relevant manuscripts indicating a role for insula in social behavior namely the Rogers-Carter et al 2018 paper that is mentioned in the introduction and also Gerlach et al 2019. 

The discussion should begin with a concise account of the key findings. Instead it starts with discussing a relatively small group of social-off cells and then moves onto very speculative points about where these neurons might project in hypothetical circuits. For example, this sentence “The Social-ON and Social-OFF ensembles can potentially increase signal-to-noise ratio of information coding at a population level and can separately route opposing information via segregated projections to distinct downstream neurons.” Is completely speculative and not supported by the provided data or set in a discussion context that is meaningful (concepts like S/N & projection circuits are not introduced). 

Line 228 of the discussion is possible incorrect. Insula has relatively few projects to PL in rat (the papers cited are rat, not mouse papers) with a heavy projection to IL. A quick pass at the Allen connectivity atlas for mouse dose show PL projections, but they are more sparse than ventral PFC areas and depend highly on the location of the tracer. The point here is that there is a lot of speculation about the circuits between AI and other regions, but the reviewed literature may be misrepresented, and the reported data offer no support for these track specific issues. 

To address this, the authors should spend more time considering what type of information would be encoded by the different subsets of cells and what other findings about insular cortex these can relate to. The second paragraph of the discussion begins to do this, but again veers into speculation. Something that the authors might consider framing their cells against findings of others, for example, Gerhlach et al show clear role of this region in fear and anxiety expression, Rogers-Carter et al 2018, and recently 2019, show effects only on social emotional behavior but not sociality per se. In the third paragraph there discussion returns to a theme set up in the introduction regarding the salience network. It would be nice to see more analyses and discussion of how the neurons found in the insula might correlate with “salience” within this context. 

Overall, these data are important, and it is worth the effort to revise this manuscript so that the key conclusions are more accessible to the reader and the discussion is focused on the inferences that can be drawn directly from the data rather than speculation. I also suggest the authors consider including the following recent publications that relate to this topic: 

1: Rogers-Carter MM, Djerdjaj A, Gribbons KB, Varela JA, Christianson JP. Insular

Cortex Projections to Nucleus Accumbens Core Mediate Social Approach to Stressed 

Juvenile Rats. J Neurosci. 2019 Oct 30;39(44):8717-8729. doi:

10.1523/JNEUROSCI.0316-19.2019. Epub 2019 Oct 7. PubMed PMID: 31591155; PubMed

Central PMCID: PMC6820210.

2: Rogers-Carter MM, Christianson JP. An insular view of the social

decision-making network. Neurosci Biobehav Rev. 2019 Aug;103:119-132. doi:

10.1016/j.neubiorev.2019.06.005. Epub 2019 Jun 10. Review. PubMed PMID: 31194999;

PubMed Central PMCID: PMC6699879.

3: Gehrlach DA, Dolensek N, Klein AS, Roy Chowdhury R, Matthys A, Junghänel M,

Gaitanos TN, Podgornik A, Black TD, Reddy Vaka N, Conzelmann KK, Gogolla N.

Aversive state processing in the posterior insular cortex. Nat Neurosci. 2019

Sep;22(9):1424-1437. doi: 10.1038/s41593-019-0469-1. Epub 2019 Aug 27. PubMed

PMID: 31455886.

4: Berret E, Kintscher M, Palchaudhuri S, Tang W, Osypenko D, Kochubey O,

Schneggenburger R. Insular cortex processes aversive somatosensory information

and is crucial for threat learning. Science. 2019 May 31;364(6443). pii:

eaaw0474. doi: 10.1126/science.aaw0474. Epub 2019 May 16. PubMed PMID: 31097492.

Reviewer #3: The authors have discovered Social-ON and Social-OFF cells intermingled in the agranular insular cortex (AI) and they showed that the Social-ON cells represented social experience independent of location and consisted of multiple subsets, each of which were preferentially active during exploration under particular behavioral state or with a particular target of physical contact.

Using a direct interaction they identified Social-ON cells (about 20% of total) and Social-OFF cells (about 3%), with some Social-ON cells exhibiting activity in specific parts of social interaction – such as stationary or moving, or contacting the nose, body or anogenital area.

Then they used a modified 3-chamber test to show again B(S)-ON cells (16%) more active in social nose poke and B(S)-OFF (4%) less active in social nose poke, with some overlap between the Social-ON vs B(S)-ON cells and Social-OFF vs B(S)-OFF cells.

In general the finding of the AI cells with activity is interesting, though the study seems somewhat preliminary and the second set of experiments with the 3-chamber test does not seem to add much to the initial finding. For example, it would have been more interesting to test whether female stimulus evokes similar response and whether those cells are overlapping. Also the choice to test AI is not particularly well justified and it would be interesting if similar cells are in other related cortices or if this is a unique feature to AI. Finally, multi-color retrograde tracers could have been used to test whether these cells have some common projections, which would strongly support their unique function.

---

## [Decision Letter · Decision Letter 2]

26 Jun 2020

Dear Dr Takumi,

Thank you for submitting your revised Short Report entitled "Encoding of social exploration by neural ensembles in the insular cortex" for publication in PLOS Biology. I have now obtained advice from the original reviewers and have discussed their comments with the Academic Editor. 

Based on the reviews, we will probably accept this manuscript for publication, assuming that you will modify the manuscript to address the remaining points raised by reviewer 2. Please also make sure to address the data and other policy-related requests noted at the end of this email.

We expect to receive your revised manuscript within two weeks. Your revisions should address the specific points made by reviewer 2. Please submit the following files along with your revised manuscript:

In addition to the remaining revisions and before we will be able to formally accept your manuscript and consider it "in press", we also need to ensure that your article conforms to our guidelines. A member of our team will be in touch shortly with a set of requests. As we can't proceed until these requirements are met, your swift response will help prevent delays to publication.

*Copyediting*

*Published Peer Review History*

*Early Version*

*Submitting Your Revision*

Sincerely,

Gabriel Gasque, Ph.D., 

Senior Editor

PLOS Biology

ETHICS STATEMENT:

>> Please indicate within the manuscript, explicitly, the name of the national or international ethical guidelines to which your experimental protocols approved by the RIKEN Animal Experiments Committee adhered. 

>> Please provide within the manuscript the ID number of the protocols approved by the RIKEN Animal Experiments Committee.

FINANCIAL DISCLOSURE

>> Please update your financial disclosure in the submission system to indicate the sources of funding that *directly* supported this work.

DATA POLICY:

>> Please update your S1 Data File to include data for figure 2F and S1B. 

>> Please ensure that the figure legends in your manuscript include information on where the underlying data can be found.

Reviewer remarks:

Reviewer #1: This manuscript by Miura et al. has been improved by the authors revisions. The authors have addressed the concerns I raised during the first round of review. I have no further comments on the manuscript.

Reviewer #2: In this revised manuscript the authors have improved from the original but the report still has room for improvement. Briefly, the authors used miniscopes and calcium imaging in the anterior insular cortex of mice performing either a social interaction task, or a linear box sociability task with a novel object or a novel conspecific at either end. In the social interaction test, ~22% of recorded cells increased firing to social interaction and of these cells some exhibited preferential firing for specific conspecific body parts. This is probably the most exciting and new piece of information in the report. In the linear sociability task, the largest portion of cells identified increased firing to the novel mouse, but only a few (~3%) of these cells were consistently "social." The authors make a number of inferences from these observations about the nature of social encoding in anterior insula. While the inferences may eventually be determined accurate, at this point, without mechanistic tests (i.e. loss of function study of these specific cells) the authors must broaden the range of inferences and specifically suggest alternatives to a strictly social account. It seems that the manuscript, and perhaps, some of the author's inferences could be tested with additional graph analysis of the putative neuronal ensembles discovered.

Ln 88. It is distracting to mention the small fraction of movement and stationary cells and then return to discuss the Social-ON cells that correlated with specific interaction types. In this reviewer's opinion, they should be moved to a supplementary section or not mentioned at all as they represent such a small fraction of the recorded cells (about 5%).

Ln 107. The support for this statement is not robust. The authors appear to be using that some cells that increase firing during social interaction also encode behavioral state and target of contact. I suggest tempering this inference or conducting more analyses to establish "conjunctive information." All of the categories are defined by human observers, what the firing actually encodes to the mouse could be quite different (and may not be "conjunctive").

Ln 178. Only 3.6% of cells initially defined as Social cells were consistent across sessions. The authors seem to interpret this as a significant amount. However, if the insula is actually a salience detector and sight of sensory integration, one might actually predict that cells don't have unitary types of encoding that would be seen in sensory systems or hippocampal place cells. Simply tallying these groups of cells seems is an underpowered approach to analysis and so the inferences made may also be too simplistic. Did the authors consider functional connectivity/graph network analysis of activity among cells. A fascinating possibility is that over sessions, the same sort of information (social or not) is encoded within the insula, but the specific cells that carry such information changes. This occurs in the circuits that orchestrate birdsong—while bird songs are consistent, the cells that "sing" the song are networked together such that a given note (or bit of information) may be played by any one of several interconnected neurons that, once fired, will trigger the next network node and the song continues. Given the vast number of circumstances in which insular cortex increases activity (across rodent and human studies) it seems more likely that these social cells are actually a sign of some other type of encoding (salience is one possibility) that is not specific to task, but might change depending on internal states, past experience, etc. For the amount of data available from the miniscope investigation, the analysis and working model that the investigators present remains under developed. 

Ln 210. This inference is one of many (that insula neurons encode the status of social exploration). An alternative account already has more support in the literature which is that insula encodes the internal states/values (see Linveh et al recent papers). It is likely that as social interaction progresses, the focal test mouse state changes. There is a challenge to interpreting social cells because it is very difficult to untangle whether the cells encode the "other" or if they encode something evoked by the other within the "self." 

Reviewer #3: I have no additional comments for the authors

---

## [Editor Report · Decision Letter 3]

24 Aug 2020

Dear Dr Takumi,

On behalf of my colleagues and the Academic Editor, Eunjoon Kim, I am pleased to inform you that we will be delighted to publish your Short Reports in PLOS Biology. 

Early Version

PRESS 

Kind regards,

Alice Musson

Publishing Editor, 

PLOS Biology

on behalf of

Gabriel Gasque,

Senior Editor

PLOS Biology